# Olive Pomace Oil versus High Oleic Sunflower Oil and Sunflower Oil: A Comparative Study in Healthy and Cardiovascular Risk Humans

**DOI:** 10.3390/foods11152186

**Published:** 2022-07-22

**Authors:** Susana González-Rámila, Raquel Mateos, Joaquín García-Cordero, Miguel A. Seguido, Laura Bravo-Clemente, Beatriz Sarriá

**Affiliations:** Department of Metabolism and Nutrition, Institute of Food Science, Technology and Nutrition (ICTAN-CSIC), Spanish National Research Council (CSIC), José Antonio Nováis 10, 28040 Madrid, Spain; s.gonzalez@ictan.csic.es (S.G.-R.); raquel.mateos@ictan.csic.es (R.M.); j.garcia@ictan.csic.es (J.G.-C.); m.seguido@ictan.csic.es (M.A.S.); lbravo@ictan.csic.es (L.B.-C.)

**Keywords:** olive pomace oil, high oleic sunflower oil, sunflower oil, lipid profile, lipid peroxidation, anthropometric parameters, clinical trial

## Abstract

Olive pomace oil (OPO) is mainly a source of monounsaturated fat together with a wide variety of bioactive compounds, such as triterpenic acids and dialcohols, squalene, tocopherols, sterols and aliphatic fatty alcohols. To date, two long-term intervention studies have evaluated OPO’s health effects in comparison with high oleic sunflower oil (HOSO, study-1) and sunflower oil (SO study-2) in healthy and cardiovascular risk subjects. The present study integrates the health effects observed with the three oils. Two randomized, blinded, cross-over controlled clinical trials were carried out in 65 normocholesterolemic and 67 moderately hypercholesterolemic subjects. Each study lasted fourteen weeks, with two four-week intervention phases (OPO versus HOSO or SO), each preceded by a three-week run-in or washout period. Regular OPO consumption reduced total cholesterol (*p* = 0.017) and LDL cholesterol (*p* = 0.018) levels as well as waist circumference (*p* = 0.026), and only within the healthy group did malondialdehyde (*p* = 0.004) levels decrease after OPO intake versus HOSO. Contrarily, after the SO intervention, apolipoprotein (APO) B (*p* < 0.001) and Apo B/Apo A ratio (*p* < 0.001) increased, and to a lower extent APO B increased with OPO. There were no differences between the study groups. OPO intake may improve cardiometabolic risk, particularly through reducing cholesterol-related parameters and waist circumference in healthy and hypercholesterolemic subjects.

## 1. Introduction

The content and quality of dietary fat is one of the most controversial issues in the field of nutrition and dietetics. National and international nutritional guidelines and recommendations share similar figures for total dietary fat intake, i.e., between 20–35% of daily energy for the general population [1,2]. However, the PREDIMED (PREvention with MEDiterranean Diet) study opened new avenues by proposing that total fat intake may cover up to 40.5% of daily energy, provided that olive oil is one of the main sources (20–25%, 45–55 mL/d) [3,4]. Virgin and extra virgin olive oil (VOO and EVOO, respectively) have well established nutritional benefits and health effects, especially against cardiovascular diseases (CVDs), which have been extensively demonstrated in clinical studies [5,6,7]. These properties have been attributed to the high content in monounsaturated fatty acids (MUFA), as well as valuable minor components such as phenols, phytosterols, tocopherols and squalene [7,8]. In contrast, the health effects of other categories of olive oil, such as olive pomace oil (OPO), have been less studied.

Olive oil extraction generates huge quantities of byproducts, and thus has severe environmental impact. Considering that OPO is obtained from the product that remains after VOO has been mechanically extracted [9,10], the consumption of this oil fosters the sustainability of the olive oil chain and promotes circular economy. OPO contains a high percentage of oleic acid (56–85% of the total fatty acids) and a wide variety of bioactive compounds, such as triterpenic acids and dialcohols, squalene, tocopherols, sterols, aliphatic fatty alcohols and phenolic compounds [11,12,13]. As a consequence of the extraction and refining processes carried out to obtain this oil, OPO differs from VOO in the content of certain minor components, lacking (poly)phenols, but then it contains higher concentrations of triterpenic compounds (mainly pentacyclic terpenic acids and diols) and aliphatic fatty alcohols than VOO [10,13]. In vivo or preclinical studies carried out with some of the above compounds have demonstrated cardioprotective properties through the modulation of lipid profile, improvement of endothelial function, induction of antihypertensive effects, reduction of inflammation levels and improvement of biomarkers related to diabetes and obesity prevention [10,12,14,15,16].

Due to OPO’s organoleptic properties, particularly its mild taste, compared to other categories of olive oil, it is more suitable for frying, and thus it is mainly demanded as a “frying oil”. Sunflower oil (SO) is also commonly used as a frying oil due to its neutral taste. In contrast to OPO, SO is a polyunsaturated fat with a high linoleic acid content that is consumed worldwide due to its frying performance, accessibility and price [17]. The growing number of claims in favor of olive oil consumption led the food industry to introduce high oleic sunflower oil (HOSO; with an oleic acid content similar to olive oil) obtained by selection or genetic modification that additionally has shown high thermal stability [18]. OPO’s frying behavior has recently been evaluated in a study conducted by Holgado et al. [19] in which OPO showed better discontinuous frying performance compared to SO and HOSO. Unlike OPO, SO’s health properties have been evaluated in numerous clinical trials in comparison with other fat sources such as coconut oil [20], canola oil, olive oil [21] and flaxseed oil [22]. Among these studies, only in the intervention by Akrami et al. [22], after the consumption of 25 mL/d of SO for 7 weeks, beneficial effects on lipid profile, blood pressure and anthropometric measures were described. On the other hand, already in 2007, the beneficial effects of HOSO on the concentrations of total cholesterol (T-C), low-density lipoprotein cholesterol (LDL-C) and high-density lipoprotein cholesterol (HDL-C) were observed after the consumption of 3.9 g/d of this vegetable oil for 12 weeks [23].

Bearing in mind the lack of human studies with OPO and its great health potential due to its composition, the aim of this research was to assess the effects of regular intake of OPO compared to the widely used SO and the nutritionally improved HOSO on biomarkers of cardiovascular health and associated pathologies (hypertension, diabetes and obesity) in healthy volunteers and subjects at cardiovascular risk. For this purpose, two randomized, blinded, cross-over controlled clinical trials, in which OPO was compared with HOSO (Study 1) and SO (Study 2), were integrated.

## 2. Materials and Methods

### 2.1. Chemical Characterization of the Study Oils

The oils used in the study were analyzed according to the following standardized methods: ISO 12228-2:2014 method for the determination of sterols; Regulation (EEC) No. 2568/91 Annex V for determining triterpenic alcohols; Regulation (EEC) No. 2568/91 Annex XIX for determining aliphatic alcohols; Regulation (EEC) No. 2568/91 Annex X for determining fatty acid composition; and ISO 9936:2016 for determining tocopherols and tocotrienols. Triterpenic acids were analyzed following the method of Pérez-Camino & Cert (1999) [24], and squalene was determined by gas chromatography (Giacometti, 2001) [25]. Phenols were analyzed by high-performance liquid chromatography with on-line diode array detection (HPLC-DAD) according to the procedure by Mateos et al. (2001) [26].

### 2.2. Study Design and Intervention

In the present study, the data obtained from the only two randomized, blinded, crossover, controlled clinical trials developed with OPO are integrated. Each study lasted 14 weeks and consisted of two four-week intervention phases, preceded by a three-week run-in or a washout period. In the first study, after a three-week run-in period with SO, half of the volunteers of each study group consumed OPO, and the other half consumed HOSO for four weeks. Then, after a three-week washout period with SO, participants consumed the other oil (OPO or HOSO) for four weeks (Figure 1). Regarding the second study, the same study design was applied, but instead, OPO was compared to SO and corn oil (CO) was used as the run-in/washout oil (Figure 1).

Participants attended the Human Nutrition Unit (HNU) of the Institute of Food Science, Technology and Nutrition (ICTAN) before and after each stage; therefore, five visits were made in both studies (Figure 1). At each visit, volunteers arrived at the HNU after an overnight fast and provided a sample of their first morning urine and the 72 h food intake record they had completed. Subsequently, blood samples were collected and blood pressure and body composition were measured. Before leaving, the volunteers were provided with the study oil according to the stage of the study. Participants were instructed to substitute the oil they usually consumed with 45 g/d (equivalent to 4–5 tablespoons) of the study oil (OPO, HOSO or SO for Study 1, and OPO, SO or CO for Study 2) to cover 20% of the daily energy intake of monounsaturated fats (equivalent to 44–67 g/d for 2000–3000 Kcal/d, respectively) following the Spanish Society of Community Nutrition (Sociedad Española de Nutrición Comunitaria, SENC) recommendations. In addition, they had to maintain their lifestyle and dietary habits, except from certain food rich in mono- and polyunsaturated fats (olives, sunflower seeds, nuts, avocado, margarine, butter and mayonnaise), which were restricted. Participants received more oil than necessary (1 L/week) to avoid the use of other oils in culinary preparations for the family; for this reason, participants did not return the oil at the end of each stage.

### 2.3. Subjects

Volunteers were recruited mainly from a database of volunteers who participated in previous studies carried out in ICTAN by placing advertisements at the faculties of the Complutense University of Madrid, and through social networks. Men and women aged between 16 and 55 years with a body mass index (BMI) less than 30 kg/m^2^ who met the following inclusion criteria were recruited; they were required to not be under medical treatment, not suffer from pathologies or chronic disorders other than moderate hypercholesterolemia in subjects at cardiovascular risk, not be smokers, not be vegetarians, not be pregnant women, and not take vitamins or dietary supplements. Participants were divided into two groups according to their T-C and LDL-C levels, which were determined in a consented blood test: normocholesterolemic subjects (T-C < 200 mg/dL; LDL-C < 135 mg/dL) and hypercholesterolemic subjects (T-C 200–300 mg/dL; LDL-C 135–175 mg/dL). Studies were conducted according to the guidelines laid down in the Declaration of Helsinki, and all procedures involving human subjects were approved by the Clinical Research Ethics Committee of Hospital Universitario Puerta de Hierro, Majadahonda in Madrid (Spain) and the Bioethics Committee of Consejo Superior de Investigaciones Científicas (CSIC). Written informed consent was obtained from all subjects. The studies were registered in Clinical Trials: Study 1 (NCT04997122) and Study 2 (NCT04998695).

### 2.4. Randomization and Blinding

All participants who met the inclusion criteria were randomly assigned to start with OPO or the control oil (HOSO for Study 1, and SO for Study 2) in a 1:1 ratio. Assignment of codes to participants, randomization and allocation to each oil were performed by different members of the research team using Microsoft**^®^** Excel 2016 software version 16.0 (Washington, DC, USA). According to T-C and LDL-C levels, two groups (normo- and hypercholesterolemic) were established regardless of sex. To blind the volunteers, the oils were presented in the same type of bottle, although with different color caps. The correspondence between the color and the study oil was known only by members of the research team.

### 2.5. Dietary Control and Compliance

The data obtained from the 72 h food intake records, which were filled out at each stage of the study, were processed using the DIAL*^®^* program (Faculty of Pharmacy-Universidad Complutense de Madrid and Alce Ingeniería). Thus, intake of energy, macronutrients (proteins, fats and carbohydrates), micronutrients (vitamins and minerals) and dietary fiber were obtained. In this record, volunteers were instructed to indicate as accurately as possible the ingredients and quantities of food consumed. When a scale was available, they were asked to record the weight of the portions consumed; otherwise, a practical manual (Manual of Nutrition and Dietetics, UCM) was provided to facilitate the interpretation of the weights and usual portions. Compliance regarding test oil intake and dietary restrictions was monitored by weekly calling and emailing volunteers.

### 2.6. Lipid Profile and Endothelial Biomarker Analysis

At each visit, a fasting blood sample was drawn in blood tubes without anticoagulant or EDTA-coated to obtain serum and plasma samples, respectively. After separation by centrifugation, the samples were frozen at −80 °C until analysis. T-C, triglycerides (TG) and HDL-C levels, as well as apolipoproteins A1 (Apo A1) and B (Apo B) were determined in serum samples following reference methods or methods recommended by Sociedad Española de Bioquímica Clínica y Patología Molecular (SEQC) using a Roche Cobas Integra 400 plus analyzer (Roche Diagnostics, Mannheim, Germany). LDL and VLDL (very low-density lipoprotein) were calculated according to the Friedewald formula—LDL = T-C − (HDL-C + TG)]; VLDL = TG/5 [27]—and Apo B/Apo A1, LDL/HDL and T-C/HDL ratios were calculated.

Regarding endothelial function biomarkers, endothelial nitric oxide synthase (eNOS, SEA868Hu), E-selectin (SEA029Hu) and P-selectin (SEA569Hu) plasma concentrations were determined in duplicate by ELISA (Cloud-Clone Kit Corp., Katy, TX, USA) using a Bio-Tek*^®^* Synergy™ HT Multi-Detection Microplate Reader controlled by BioTek*^®^*Gen5 version 2.01.14 software (BioTek Instruments, Winooski, VT, USA).

### 2.7. Blood Pressure and Anthropometric Measurements

At the end of each stage, after volunteers had rested for 5 min, systolic (SBP) and diastolic (DBP) blood pressure was measured in triplicate using an OMRON**^®^** M2 HEM-7121-180 E sphygmomanometer (OMRON HEALTHCARE Co., Ltd., Kyoto, Japan), waiting for 3 min between measurements.

In order to determine the effect of OPO, HOSO and SO consumption on body composition, height and body circumferences (waist and abdomen) were measured using a height rod (Soehnle Professional, GmBH) and a measuring tape (Fisaude ADE, Spain), respectively. In addition, weight and the percentage of body fat were estimated using a single-frequency tetrapolar electrical bioimpedance using a Tanita**^®^** BC 601 segmented body composition analyzer which includes a digital scale (Tanita Europe BV, Amsterdam The Netherlands). Waist to hip and waist to height ratios were calculated.

### 2.8. Analysis of Antioxidant Capacity and Oxidation Biomarkers

Antioxidant activity was measured in serum samples by the ABTS radical cation [28] and the oxygen radical absorbance capacity (ORAC) methods [29], and the reducing capacity was determined by the ferric reducing/antioxidant power (FRAP) assay [30]. Trolox was used as a standard and results were expressed as μM of Trolox equivalent (TE). Low-density lipoprotein oxidation (LDLox) levels were determined in serum samples by ELISA assay according to the protocols attached in Cloud-Clone Corp. kit (Katy, TX, USA). These parameters were analyzed using a Bio-Tek**^®^** Synergy™ HT Multi-Detection plate reader (Highland Park, Winooski, VT, USA) controlled by BioTek**^®^**Gen5 software version 2.01.14.

Malondialdehyde (MDA) levels were measured as a biomarker of lipid oxidation. MDA was determined in serum samples by high-performance liquid chromatography (HPLC) following the methodology proposed by Mateos et al. [31]. For this purpose, a 1200 series HPLC equipment (Agilent Technologies, Santa Clara, CA, USA) and a Nucleosil 120 C18 column (25 mm × 0.46 mm, particle size 5 µm, TeknoKroma) were used.

### 2.9. Statistical Methods

The study design took the group (normocholesterolemic/hypercholesterolemic) and treatment (OPO, HOSO or SO) (repeated measures) as fixed factors and the order of intake as a random factor. In addition, to control for possible variability between the two populations, the study (Study 1 or Study 2) was also taken as a random factor (without repeated measures).

The statistical models applied to analyze the results of this study are presented below:A general linear repeated measures model was used to analyze energy, macronutrient and micronutrient intakes over the course of each study, so that values at baseline, initial (pre-treatment) and final (post-treatment) results were compared. The order of intake was not taken into account. Results are shown as mean ± standard error of the mean.A linear mixed model was applied to study the rate of change [(post-treatment value—pre-treatment value)/pre-treatment value] of each variable within each group (normocholesterolemic or hypercholesterolemic). This statistical model takes into account the correlated and non-constant variability of the data. Thus, it is possible to contemplate the order of oil intake. The statistical model was full factorial, considering group (normocholesterolemic or hypercholesterolemic), treatment oil (OPO, HOSO or SO) and group*treatment interaction, while the order of intake was a random effect. Results are expressed in percentage as mean ± standard error of the mean.

Data were analyzed using SPSS software (version 27.0; SPSS, Inc., IBM Company New York, NY, USA). Prior to the aforementioned statistical analysis, the data distribution was tested for normality using the Kolmogorov–Smirnov test. The box plot was applied to check the distribution of all variables. Furthermore, the Bonferroni test (within each group) was performed to pairwise compare the effect of the intake of each oil (OPO, HOSO and SO). The significance level was set at *p* < 0.05.

## 3. Results and Discussion

VOO and EVOO are extracted under conditions which allow the oil to retain distinctive flavor compounds from the olives [19]. Thus, although the nutritional and health properties of VOO and EVOO are unquestionable, the bitter aftertaste may make them less suitable for cooking and in addition, their cost is higher than other seed oils, particularly in countries with no olive oil production [17]. Bearing this in mind, OPO, due to its mild flavor and cooking properties, is an interesting alternative to HOSO and SO, which are consumed worldwide [19]. Moreover, OPO has relevant health potential according to in vitro and preclinical studies carried out mainly using OPO, enriched OPO or OPO components [13]. However, to date, no free-living, long-term intervention has been carried out in healthy and cardiovascular risk humans replacing the oil usually consumed with OPO in the context of their habitual diet to better understand OPO’s effects on cardiovascular health and associated comorbidities.

### 3.1. Chemical Composition of the Study Oils

The chemical characterization of the vegetable oils compared in this study, OPO, HOSO and SO, is shown in Appendix A. OPO and HOSO are two monounsaturated fat sources with a high oleic acid content, 71% and 76.5%, respectively, whereas SO is a polyunsaturated fat source due to its high linoleic acid content (58.6%). As far as minor components are concerned, OPO, HOSO and SO showed similar α-tocopherol (OPO: 357 mg/kg; HOSO: 420 mg/kg; SO: 518 mg/kg) and phenolic compounds (<2.0 mg/kg in OPO, HOSO and SO) content. Nevertheless, OPO showed a considerably higher content in squalene (799 mg/kg), aliphatic alcohols (978 mg/kg), triterpenic alcohols (886.6 mg/kg) and triterpenic acids (196 mg/kg) compared to the low content or absence in HOSO (87 mg/kg squalene; 32 mg/kg aliphatic alcohols; <2.0 mg/kg triterpenic alcohols and acids) and SO (117 mg/kg squalene; 26 mg/kg aliphatic alcohols; <2.0 mg/kg triterpenic alcohols and acids). OPO’s chemical composition was within the ranges reported in the review by Mateos et al. [13], with the exception of alpha tocopherol (357 mg/kg), which was slightly above the range of 185–300 mg/kg, and aliphatic fatty alcohols (978 mg/kg), which were marginally under (1000–3000 mg/kg) [13]. With respect to SO and HOSO, the fatty acid composition was similar to that determined in previous studies [19].

Regarding the oils used in the washout periods, sunflower (SO) and corn oil (CO), their composition is quite alike. Both SO and CO oil are polyunsaturated fats with a similar content in their major components: linoleic acid (58.48% and 50.52% in SO and CO, respectively) followed by oleic acid (29.76% and 35.36% in SO and CO, respectively). With respect to minor compounds (representing about 2% of the total composition), the content of triterpenic acids (<2.0%), triterpene alcohols (<1.0%) and phenols (<1.0%) were similar in both oils, whereas their content in aliphatic fatty alcohols (38 mg/kg and 29 mg/kg in SO and CO, respectively) and tocopherols (230 mg/kg and 246 mg/kg in SO and CO) differed very little. In contrast, the contents of squalene, tocotrienols and sterols were slightly higher in CO (squalene: 548 mg/kg; tocotrienols: 19 mg/kg; sterols: 8962.0 mg/kg) compared to SO (squalene: 314 ppm; tocotrienols:—mg/kg; sterols: 2820.5 mg/kg).

### 3.2. Paticipants’ Characteristics, Dietary Control and Compliance

Of the 144 volunteers recruited, 132 successfully completed both studies: 65 were normocholesterolemic and 67 hypercholesterolemic. The baseline characteristics of the subjects are shown in Table 1. As expected, variables related to lipid profile were higher in the cardiovascular risk group compared to in the healthy. Additionally, anthropometric and blood pressure measurements were higher in the hypercholesterolemic group.

The assessment of dietary energy, macronutrient and micronutrient intakes are shown in Table 2. Repeated measures analysis revealed that energy, protein, carbohydrate and lipid intake did not vary in the normo- or hypercholesterolemic group after OPO, HOSO and SO intake (Table 2), confirming that participants maintained their dietary habits throughout the study as requested and replaced the oil usually consumed with the test oil. To assess energy and nutrient intake, the recommended daily intakes reported by Moreiras et al. [32] were taken as reference values. Regarding total caloric intake (Table 2), mean values ranged from 1966 to 2261 kcal/d, which are slightly under the recommended for healthy adults (2000–3000 kcal/d). This result is not a surprise, because although volunteers were instructed on how to fill in the dietary records, there is a widespread tendency to underestimate the amount of food consumed. Considering this drawback, regarding macronutrient intake, the distribution of protein (between 80–98 g/d), carbohydrate (between 175–214 g/d) and lipid (between 88–102 g/d) intake accounted for 17%, 38% and 40.6%, respectively (Table 2). These values were not within the recommended ranges of 10–15%, 50–60% and 30–35% for protein, carbohydrate and lipid intake, respectively, showing higher protein and lipid intake at the expense of lower carbohydrate intake. These results agree with previous studies that point to diets in Mediterranean countries moving away from the Mediterranean dietary pattern [33,34], with a recurrent tendency to consume more proteins and lipids and less carbohydrates as observed in other studies [35].

Consumption of SFA (between 24–31 g/d), MUFA (between 28–46 g/d) and polyunsaturated fatty acids (PUFA) (between 10.3–33.2 g/d) accounted for 12.3%, 15.6% and 9.5% (obtained from mean values) of total dietary energy, respectively. SFA intake (12.3% of the total diet) showed no variation in either group and was above the total recommended intake (<7%). As for MUFA consumption (15.6% of the total diet), it was close to the recommended intake (17%) and, as expected, it showed significant changes (*p* < 0.001) in both the normocholesterolemic and hypercholesterolemic groups due to the treatment oil (OPO, HOSO and SO) (Table 2). According to the Bonferroni test, MUFA consumption was similar at baseline and after OPO and HOSO intake (Table 2), which may be related to the fact that volunteers usually consumed VOO, with a high content in MUFA, similar to OPO and HOSO, both rich in MUFA. However, the mean intake of PUFA (9.5% of the total diet) was above the 3–6% recommended intake. This parameter also experienced variations after OPO, HOSO and SO intake (*p* < 0.001) in both groups, which reflected the different contents in PUFA in the test oils, so that it increased after SO intake, rich in polyunsaturated fatty acids, and decreased after OPO and HOSO intervention (Table 2). With respect to total cholesterol intake, which was slightly above the 300 mg/d recommendation, there were no significant differences due to dietary intervention. There were also no differences in the intake of dietary fiber, which remained close to the lower limit recommended (25–35 g/d; Table 2). Finally, vitamin E consumption showed statistically significant variations associated with the consumption of the different oils (*p* < 0.001) in both groups. Baseline intake (9.1–9.5 mg/d) was below the recommended 12 mg/d [36]; however, the intake increased significantly after OPO, HOSO and SO consumption (15.4–28.1 mg/d; Table 2), without differences among the study oils, due to the higher content in vitamin E of these oils compared to VOO, which was the habitual oil consumed by volunteers.

### 3.3. Lipid Profile

The effects of regularly consuming OPO, HOSO and SO on the volunteer’s lipid profile are shown in Appendix A. The changes in T-C (*p* = 0.017), LDL-C (*p* = 0.018), Apo B (*p* < 0.001) and Apo B/Apo A1 (*p* < 0.001) ratio (Figure 2) induced by the oils studied were statistically significant, whereas the effects of the study groups (normocholesterolemic and hypercholesterolemic) were not in any the parameters studied. It is noteworthy that, after sustained OPO consumption, the change rate of T-C and LDL-C levels were negative, thus indicating that these parameters decreased at the end of the intervention compared to the beginning. In contrast, after regularly consuming HOSO and SO, T-C and LDL-C slightly increased, except for T-C in the risk group after SO ingestion (−0.1% ± 1.6%; Appendix A). The potential cardioprotective effects of OPO on T-C and LDL-C are reinforced with the decrease in the LDL-C/HDL-C ratio observed after OPO intake, which was close to the level of statistical significance (*p* = 0.056) (Appendix A). These results suggest that OPO may exert positive hypolipidemic effects in both healthy and cardiovascular risk subjects, in agreement with the review by Mateos et al. [13], which highlights the cardioprotective potential of OPO. These effects may be associated to OPO’s content in triterpenic acids, mainly maslinic and oleanolic acid, in the unsaponifiable fraction, as these compounds have already been related with improving blood lipid levels [10,12,16]. In agreement, in a study carried out in hyperlipidemic patients who consumed four tablets of oleanolic acid at once/three times a day (the exact quantity administered is not indicated) for 4 weeks, reduced levels of T-C, TG and LDL-C were described [37]. Similarly, in obese mice that consumed OPO with a high content in oleanolic and maslinic acids for 10 weeks, a decrease in T-C and TG levels was reported [12]. These results along with the fact that HOSO, also rich in MUFA, did not show similar effects on T-C and LDL-C concentration, sustain our hypothesis that OPO’s hypolipidemic effects can be mainly attributed to its content in minor bioactive components.

The slight increase in T-C and LDL-C concentrations following HOSO intake contrasts with the results obtained in a previous intervention study carried out in healthy subjects, where the consumption of 3.9 g/d of HOSO for 12 weeks was related to a reduction in T-C, LDL-C and HDL-C concentrations [23]. In contrast, the lack of reduction in T-C and LDL-C concentration after SO consumption is in agreement with human studies that assessed the effects of consuming SO on blood lipids in comparison with flaxseed oil [22], coconut oil [20], canola oil and olive oil [21]. According to the Regulation (EC) No 432/2012, OPO and HOSO are high monounsaturated fats, whereas SO is high polyunsaturated fat. In addition, the Regulation indicates that replacing saturated fats with unsaturated fats (both mono and polyunsaturated) in the diets contributes to the maintenance of normal cholesterol [38]. The present results would not support the claim, as the effects on T-C were different between the two monounsaturated oils (OPO and HOSO) and the polyunsaturated fat (SO), and the volunteers did not replace saturated fats with the unsaturated fats here studied, as they usually consumed VOO and in some cases SO for cooking.

Apolipoproteins are crucial components of lipoprotein particles that help to predict the risk of cardiovascular disease, as they modulate the activity of specific enzymes that act on lipoproteins, maintain the structural integrity of the lipoprotein complex, and are ligands for specific cell surface receptors to assist lipoprotein uptake [39]. Therefore, Apo B, Apo A1 and Apo B/Apo A1 ratio are considered as important predictors of CVD alongside with lipid profile biomarkers. On one hand, elevated serum Apo B levels are associated with increased arterial wall injury, as Apo B is the main apolipoprotein involved in the initiation and progression of atherosclerosis and it is present in chylomicrons, very-low density (VLDL) and intermediate density (IDL), lipoproteins, LDL-C and lipoprotein(a) [Lp(a)] particles [40]. On the other hand, Apo A1 acts as a receptor key in the transfer of phospholipids and free cholesterol from peripheral tissues. In addition, this apolipoprotein, which is the largest component of the HDL-C particle [41], transfers cholesterol to the liver and other tissues for excretion and steroidogenesis, which is related to the protective action of Apo A1 and HDL-C in cardiovascular health [39]. Consequently, high Apo A1 levels are associated with a low cardiovascular risk index, and the Apo B/Apo A1 ratio is a representative marker of the atherogenic/anti-atherogenic molecule ratio, so that an increase in the ratio is associated with cardiovascular events [42]. According to Figure 2, Apo B increased after SO intake, as well as with OPO to a lower extent (*p* < 0.001). Contrarily, after HOSO consumption, Apo B values hardly changed in the normocholesterolemic group and slightly decreased in the hypercholesterolemic group (Figure 2). The small upward trend in Apo B after OPO intake differs from that observed in a postprandial study with OPO [43]. Therefore, further studies would be necessary to clarify the controversial Apo B result observed with OPO in the present study. Regarding Apo A1, there was a positive upward trend after OPO, HOSO and SO intake, without statistical differences among the studied oils (Appendix A). In contrast, the Apo B/Apo A1 ratio showed significantly different changes (*p* < 0.001), outstanding the increase after SO consumption, versus the decrease with HOSO and OPO in the hypercholesterolemic group (Figure 2). The marked increase after SO, a polyunsaturated fat source, compared to HOSO and OPO, two monounsaturated fats, suggests that the oleic acid present in OPO and HOSO may favorably influence this marker of cardiovascular risk.

### 3.4. Blood Pressure and Biomarkers of Endothelial Function

At baseline, systolic (from 114.3 to 121.6 mmHg) (SBP) and diastolic blood pressure (DBP) results (from 74.2 to 79.5 mmHg) (Baseline characteristics; Table 1) were within the normal range [44], as well as after regular consumption of the tested oils (SBP: 114.5 to 115.8 mmHg; DBP: 78.5 to 77.2 mmHg). Moreover, no significant changes on SBP and DBP were observed with OPO, similarly to HOSO and SO (Appendix A). The OPO results here observed contrast with those in a study carried out in hypertensive rats after consuming OPO enriched with triterpenic acids (oleanolic and maslinic acids) for 8 weeks, as the animals’ high blood pressure was attenuated [10]. Since participants in the present study were normotensive subjects, the question remains whether OPO might had elicited a potential effect on hypertensive patients thanks to its content in triterpenic acids and other minor bioactive components. More studies need to be carried out to better understand the effects of OPO on blood pressure in humans.

On the other hand, numerous studies point to MUFAs in olive oil as key components responsible for the beneficial effects on endothelial function and vascular inflammation [45,46]. This is most relevant considering that the vascular endothelium is involved in regulatory functions of the cardiovascular system, as well as in vascular maintenance and homeostasis [47]. In the present study, to evaluate the effects on endothelial status of the three different vegetable oils studied, the concentration of endothelial nitric oxide synthase (eNOS), endothelial selectin (E-selectin), platelet selectin (P-selectin), intercellular adhesion molecule 1 (ICAM-1) and vascular cell adhesion molecule 1 (VCAM-1) was measured (Table 3). Regular consumption of OPO, HOSO and SO for 4 weeks did not show significant changes in any of the biomarkers of endothelial function analyzed (*p* > 0.05) (Table 3). However, regarding eNOS, there was a tendency to increase, particularly after OPO intake (change rate: 36.8 ± 15.9 and 21.1 ± 12.1 in normocholesterolemic and hypercholesterolemic groups, respectively), as well as with HOSO in the hypercholesterolemic group (37.5 ± 16.5), whereas in the healthy group the effect was smaller (10.0 ± 23.7) (Table 3). In contrast, the opposite trend was observed after the SO intervention in both study groups (Table 3). Considering that nitric oxide (NO) is a key factor in vascular regulation due to its vasodilatory action [10], these results support that the presence of MUFAs, such as oleic acid in OPO and HOSO, can positively modulate endothelial function in line with previous studies in humans carried out with the Mediterranean Diet, rich in MUFAs [45,46]. In agreement, animal studies using triterpenic acid-enriched OPO also reported improved endothelial function in both aorta and mesenteric arteries by increasing eNOS expression [10,48]. To sum up, the high oleic acid content in OPO and HOSO seems to have slightly improved endothelial function through the activation of eNOS, and it is likely that in OPO, the content in triterpenic acids might have contributed to this beneficial effect.

Alternatively, P-selectin and E-selectin are adhesion molecules mainly expressed in states of endothelial inflammation, facilitating the movement of monocytes, neutrophils and lymphocytes [49], and thus they play an important role in the pathophysiology of atherosclerosis and endothelial dysfunction [50,51]. According to Table 3, P-selectin and E-selectin values showed an upward trend after HOSO (change rate: 20.6 ± 11.2 normocholesterolemic; 22.0 ± 11.2 hypercholesterolemic) and SO (change rate: 14.4 ± 12.3 normocholesterolemic; 16.5 ± 18.7 hypercholesterolemic) intake, respectively. The increase observed in E-selectin levels, as well as the lower expression of eNOS after SO intake, suggest a lower cardioprotective effect of this oil compared to OPO. However, after the HOSO intervention, the results are somewhat controversial, since eNOS expression increased whereas E-selectin levels decreased (−32.7 ± 11.4) in the healthy subjects, which is not consistent with the increase in P-selectin observed in both groups (Table 3). Therefore, the effects of the studied oils on endothelial function are moderate and not clear with HOSO.

### 3.5. Anthropometric Parameters and Body Composition

Among the anthropometric and bioimpedance determinations carried out to assess the effects of regularly consuming OPO, HOSO and SO on body composition (Table 4), only waist circumference significantly changed (*p* = 0.026), so that a decrease was observed after OPO (−0.5 ± 0.5 normocholesterolemic; −0.8 ± 0.4 hypercholesterolemic) and SO (−0.8 ± 0.9 normocholesterolemic; −0.6 ± 0.8 hypercholesterolemic) consumption, in contrast to HOSO (1.1 ± 0.9 normocholesterolemic; 1.6 ± 1.3 hypercholesterolemic; Table 4). Considering that the waist-to-height ratio is better for predicting CVD risk [52], this ratio as well as the waist-to-hip ratio (data not shown) were calculated; however, no significant changes were observed (*p* > 0.05). The favorable effect on waist circumference following OPO intake may be related to the presence of oleanolic and maslinic acids, according to the study carried out in obese mice that lost weight when they were changed to a diet containing OPO enriched in both triterpenic acids for 10 weeks compared to a high-fat diet [12]. This association is also supported by the effects of oleanolic acid on visceral adiposity [53] as well as on body weight [54] in obese murine models. In the present study, the reduction in waist circumference was accompanied by a decreasing trend in visceral fat and weight (without reaching the significance level) in the normocholesterolemic group, and in the hypercholesterolemic to a lower extent (Table 4). Apart from the triterpenic acids, MUFA may have also played a role in these positive effects, as the two oils rich in MUFA (OPO and HOSO) produced positive effects on visceral fat in contrast to SO. This point should be further studied, as part of the present results support the study by Lambert et al. [23], who described that the consumption of 3.9 g/d HOSO for 12 weeks had no significant effect on body fat, waist circumference and waist/hip ratio in healthy subjects; however, the effects on visceral fat observed here are promising.

In relation to SO, previous clinical studies have described that dyslipidemic subjects who substituted their usual oil with SO or canola oil (also rich in PUFA), maintaining their dietary habits, showed no significant changes in weight, BMI, waist circumference and hip circumference after the SO intervention [55]. When SO was consumed as a dietary supplement (3.2 g/d), any statistically significant change in weight, body fat, waist circumference and BMI was observed, although there was a downward trend [56]. The present results match with these studies, as there were no statistical changes in the anthropometric parameters measured, except for the decrease in waist circumference (*p* = 0.026; Table 4), which may be attributed to the higher amount of SO consumed (45 g/d) compared to the latter study 3.2 g/d [56], and likely to the former [52] where the quantity of SO oil consumed was ad libitum.

### 3.6. Oxidation and Antioxidant Biomarker Analysis

The role of oxidative stress in the onset and progression of atherosclerosis and its impact on the development of cardiovascular events has been widely described, as many studies associate increased oxidative stress with atherosclerotic risk factors, such as hypertension, dyslipidemia and obesity, among others [57,58]. In the present study, ABTS, ORAC, FRAP, oxLDL and MDA were determined to assess possible changes in oxidative status after the dietary interventions. As shown in Table 5, prolonged consumption of OPO, HOSO and SO did not induce significant changes in ABTS, ORAC, FRAP and oxLDL values in both groups, except for FRAP (*p* = 0.018), which showed differences between the study groups, so that the hypercholesterolemics showed a general greater improvement compared to the normocholesterolemics. FRAP, ORAC and ABTS are three complementary methods commonly used to evaluate the antioxidant capacity of different oils, such as olive or sunflower oil, among others [59], as well as in foods naturally containing or enriched with phenolic compounds, such as hydroxytyrosol in olive oil [60]. Although these techniques have not been applied in previous clinical studies with OPO, HOSO or SO, they have been used in long-term studies carried out with other food products naturally high in antioxidants, rich in phenolic compounds, such as cocoa [61,62], green/roasted coffee [35] or in post-prandial studies with hydroxytyrosol-enriched biscuits [63]. Among these studies, only the regular consumption of the green/roasted coffee mixture, rich in phenols due to the green coffee beans, was associated with increased FRAP and ORAC levels after the intervention. Similarly, the presence of hydroxytyrosol in the enriched biscuits significantly reduced oxLDL levels at 2 and 4 h after intake compared to a control biscuit not containing phenols. Therefore, it is likely that the lack of significant changes in FRAP, ORAC, ABTS and oxLDL observed in the present study is due to the absence of (poly)phenols in the study oils and to the fact that the minor compounds present in the oils did not produce changes in oxidation status.

Nevertheless, the biomarker of oxidative lipid damage, MDA, showed an interaction between the oil and the group (*p* = 0.004; Table 5). According to the Bonferroni test, differences were found in the normocholesterolemic group, with a decrease (−4.4 ± 2.2) and an increase (3.1 ± 3.2) after OPO and HOSO intake, respectively (Table 5). The different behavior between the two sources of monounsaturated fat indicates that certain component(s) of the unsaponifiable fraction of OPO may have exerted the beneficial effects on MDA levels observed. This idea is supported by the outcome in rats with induced myocardial infarction (using isoproterenol); after the administration of maslinic acid (15 mg/kg body weight) for 7 days, the animals showed a reduction in MDA levels [64]. However, the dose of maslinic acid in the present study (0.10 mg/d) provided by the 45 g/d of OPO is far below that used in Hussain et al. [64]. Finally, it should not be disregarded that an antioxidant component in oils with well-recognized protective effects against reactive oxygen species is α-tocopherol (vitamin E) [13]. Although the intake of this micronutrient was significantly higher after the SO intervention (Table 2), this was not associated with a greater antioxidant capacity after SO intake compared to the other two oils studied. Attending to the Commission Regulation (UE) 432/2012, the three oils evaluated in this study (OPO, HOSO and SO) may be considered as foods high in vitamin E, which is of great interest, as, apart from its essentiality, this nutrient protects cells against oxidative damage (Commission regulation (EC) No 432/2012) [38]. However, due to the similarity in vitamin E content in the study oils (Appendix A), this nutrient has not led to differences among the oils in antioxidant capacity parameters measured.

### 3.7. Strengths and Limitations

The present work has some limitations: the oils used in the run-in and wash-out phases in Study 1 (SO) and Study 2 (CO) had a similar but not identical fatty profile. Nevertheless, the chemical composition of SO and CO was quite similar (Appendix A). In order to assess whether SO and CO induced differences in initial (pre-treatment) values, we statistically analysed this data and did not observe differences. In addition, we used rates of change to minimize any possible differences in the initial values due to the use of SO or CO, and focused on the changes induced by OPO consumption. Another limitation was that, although the presentation of the oils was with the same bottle format for blinding, given the familiarity with sunflower oil, it is likely that the participants knew the type of oil they were consuming. However, important strengths are identified: the study design was robust, and a large number of parameters related to cardiovascular disease and associated comorbidities were analyzed, thus providing considerable information on the health effects of OPO, HOSO, and SO. In addition, the number of participants (*n*) was within the range established according to the power calculations

## 4. Conclusions

When comparatively studying the effects of regularly consuming OPO, HOSO and SO (45 g/d), only OPO produced beneficial effects on cardiovascular health, as T-C, LDL-C and waist circumference are reduced in both healthy and hypercholesterolemic subjects. In addition, serum lipid oxidation in the healthy group is lowered. All these findings are in line with previous in vitro or preclinical studies developed with the OPO or some of its components. However, the aforementioned positive cardiovascular effects of OPO are not supported by changes in Apo B levels in both groups and the Apo B/Apo A1 ratio in the normocholesterolemic participants. Taken together, it may be concluded that OPO is a healthier alternative than HOSO and SO for healthy and cardiovascular risk people.

## Figures and Tables

**Figure 1 foods-11-02186-f001:**
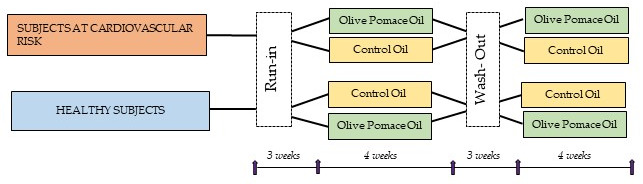
Study design. The arrows indicate the visits performed.

**Figure 2 foods-11-02186-f002:**
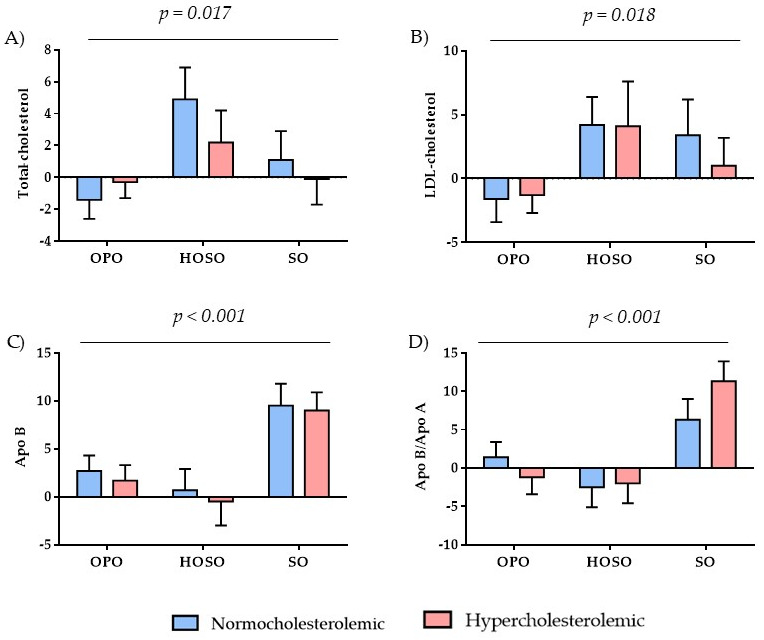
Values represent the rate of change calculated from [(post-treatment value − pre-treatment value)/pre-treatment value] and the error bars represent standard error of mean (both expressed as percentage). *n* per group: normocholesterolemic: Olive pomace oil (OPO): *n* = 65, high oleic sunflower oil (HOSO): *n* = 34, sunflower oil (SO): *n* = 31; hypercholesterolemic: OPO: *n* = 67, HOSO: *n* = 30, SO: *n* = 37. According to the linear mixed model, the oil effect was significant: (**A**) *p* = 0.017; (**B**) *p* = 0.018; (**C**) *p* < 0.001; (**D**) *p* < 0.001. There were no significant differences between the study groups, neither oil × group interaction (*p* > 0.05). Apo: apolipoprotein.

**Table 1 foods-11-02186-t001:** Baseline characteristics of participants in both studies.

	Normocholesterolemic	Hypercholesterolemic
Men, *n*	21	36
Women, *n*	44	31
Age (years)	31 ± 1	43 ± 1
BMI (kg/m^2^)	23.6 ± 0.6	25.7 ± 0.5
Waist circumference (cm)	75.6 ± 1.7	85.0 ± 1.7
Total cholesterol (mg/dL)	177.0 ± 2.8	239.2 ± 3.3
LDL cholesterol (mg/dL)	97.5 ± 2.6	149.4 ± 3.7
Systolic blood pressure (mmHg)	114.3 ± 1.5	121.6 ± 1.8
Diastolic blood pressure (mmHg)	74.2 ± 1.0	79.5 ± 1.3

Values represent mean ± standard error of mean. BMI: body mass index; LDL: low-density lipoprotein.

**Table 2 foods-11-02186-t002:** Energy intake and dietary components.

	Normocholesterolemic							*p* Value
	Baseline	OPO (*n* = 65)		HOSO (*n* = 34)		SO (*n* = 31)		
		Initial	Final	Initial	Final	Initial	Final	
Energy (kcal/day)	2078 ± 60	1966 ± 57	2017 ± 64	2198 ± 83	2087 ± 92	2017 ± 63	2006 ± 59	0.394
Proteins (g/day)	89 ± 3	82 ± 3	86 ± 3	96 ± 4	91 ± 5	87 ± 4	80 ± 3	0.132
Carbohydrates (g/day)	196 ± 7	191 ± 8	190 ± 8	209 ± 10	205 ± 10	176 ± 7	175 ± 7	0.087
Lipids (g/day)	95 ± 4	89 ± 3	92 ± 4	100 ± 5	92 ± 5	98 ± 4	100 ± 4	0.395
SFA (g/day)	31 ± 1	27 ± 1	28 ± 1	29 ± 2	27 ± 2	30 ± 1	28 ± 2	0.065
MUFA (g/day)	41 ± 2 ^a^	29 ± 1 ^b^	45 ± 2 ^a^	31 ± 2 ^b^	46 ± 3 ^a^	33 ± 1 ^b^	30 ± 2 ^b^	<0.001
PUFA (g/day)	14.5 ± 0.8 ^a^	25.1 ± 1.4 ^bd^	12.5 ± 0.6 ^ac^	30.1 ± 1.8 ^bd^	10.7 ± 0.7 ^c^	26.4 ± 1.3 ^b^	33.2 ± 1.8 ^d^	<0.001
Cholesterol (mg/day)	327 ± 16	314 ± 18	334 ± 17	365 ± 27	335 ± 41	334 ± 18	340 ± 21	0.769
Dietary fiber (g/day)	21 ± 1	20 ± 1	20 ± 1	23 ± 1	22 ± 1	20 ± 1	21 ± 1	0.821
Vitamin E (mg/day)	9.5 ± 0.6 ^a^	20.5 ± 1.3 ^b^	17.2 ± 1.0 ^c^	24.4 ± 1.9 ^b^	17.8 ± 1.3 ^c^	16.8 ± 0.9 ^c^	26.1 ± 1.6 ^b^	<0.001
	**Hypercholesterolemic**							***p* Value**
	**Baseline**	**OPO (*n* = 67)**		**HOSO (*n* = 30)**		**SO (*n* = 37)**		
		**Initial**	**Final**	**Initial**	**Final**	**Initial**	**Final**	
Energy (kcal/day)	2099 ± 52	2146 ± 61	2079 ± 56	2261 ± 95	2084 ± 106	2006 ± 93	2027 ± 97	0.158
Proteins (g/day)	93 ± 3	91 ± 3	90 ± 3	98 ± 4	89 ± 5	84 ± 5	83 ± 4	0.121
Carbohydrates (g/day)	201 ± 8	206 ± 8	200 ± 7	214 ± 12	212 ± 16	200 ± 12	197 ± 13	0.721
Lipids (g/day)	91 ± 3	95 ± 3	91 ± 4	102 ± 6	88 ± 6	88 ± 5	91 ± 5	0.122
SFA (g/day)	29 ±1	29 ± 1	28 ± 1	30 ± 2	24 ± 2	28 ± 2	28 ± 2	0.223
MUFA (g/day)	41 ± 2 ^a^	31 ± 1 ^b^	44 ± 2 ^a^	32 ± 2 ^b^	46 ± 4 ^a^	29 ± 2 ^b^	28 ± 2 ^b^	<0.001
PUFA (g/day)	12.8 ± 0.5 ^a^	26.9 ± 1.4 ^b^	12.9 ±0.6 ^a^	30.5 ± 2.7 ^b^	10.3 ± 0.6 ^a^	23.5 ± 1.6 ^b^	26.2 ± 1.9 ^b^	<0.001
Cholesterol (mg/day)	358 ± 17	320 ± 20	346 ± 22	334 ± 25	308 ± 29	293 ± 22	354 ± 31	0.298
Dietary fiber (g/day)	24 ± 2	24 ± 1	23 ± 1	28 ± 2	23 ± 2	20 ± 2	22 ± 2	0.230
Vitamin E (mg/day)	9.1 ± 0.5 ^a^	20.8 ± 1.5 ^b^	19.8 ± 0.9 ^c^	28.1 ± 2.7 ^b^	17.7 ± 1.5 ^c^	15.4 ± 1.2 ^c^	20.9 ± 1.5 ^b^	<0.001

Values represent mean ± standard error of mean. Data were analysed using a general linear repeated measures model. Superscripts correspond to significant differences within the normocholesterolemic (N) or hypercholesterolemic (H) group according to the Bonferroni test. *p* values correspond to the comparing baseline, initial and final effects of olive pomace oil (OPO), high oleic sunflower oil (HOSO) and sunflower oil (SO). Significance level was set at *p* < 0.05. SFA: saturated fatty acid. MUFA: monounsaturated fatty acid. PUFA: polyunsaturated fatty acid.

**Table 3 foods-11-02186-t003:** Effects of olive pomace oil (OPO), high oleic sunflower oil (HOSO) and sunflower oil (SO) on endothelial function biomarkers.

(%)	Normocholesterolemic			Hypercholesterolemic			*p* Value		
	OPO *n* = 65	HOSO *n* = 34	SO *n* = 31	OPO *n* = 67	HOSO *n* = 30	SO *n* = 37	Oil	N/H	Oil × N/H
**eNOS**	36.8 ± 15.9	10.0 ± 23.7	−6.7 ± 8.9	21.1 ± 12.1	37.5 ± 16.5	−4.8 ± 11.9	0.100	0.430	0.298
**E-selectin**	6.3 ± 9.9	−32.7 ± 11.4	14.4 ± 12.3	−1.0 ± 7.8	−2.0 ± 16.8	16.5 ± 18.7	0.789	0.907	0.378
**P-selectin**	9.2 ± 5.6	20.6 ± 11.2	6.9 ± 5.1	4.1 ± 5.2	22.0 ± 11.2	2.9 ± 3.6	0.726	0.559	0.997
**ICAM-1**	−0.8 ± 3.4	2.1 ± 3.6	−1.8 ± 4.4	8.4 ± 5.6	−1.3 ± 6.9	−1.9 ± 5.2	0.241	0.513	0.351
**VCAM-1**	4.5 ± 4.3	6.8 ± 6.9	10.5 ± 8.2	18.0 ± 6.5	0.3 ± 9.9	6.1 ± 9.4	0.627	0.802	0.158

Values represent mean ± standard error of mean. The rate of change was calculated from [(post-treatment value − pre-treatment value)/pre-treatment value] and expressed as percentage. A linear mixed model was used for data analysis. *p*-value in the first column represents the treatment effect (OPO, HOSO or SO), in the second column the group effect [normocholesterolemic (N) and hypercholesterolemic (H)], and in the last column the interaction of treatment and group. Significance level was set at *p* < 0.05. eNOS: endothelial nitric oxide synthase. E-selectin: endothelial selectin. P-selectin: platelet selectin. ICAM-1: intercellular adhesion molecule 1. VCAM-1: vascular cell adhesion molecule 1.

**Table 4 foods-11-02186-t004:** Effects of olive pomace oil (OPO), high oleic sunflower oil (HOSO) and sunflower oil (SO) on anthropometric parameters and body composition.

(%)	Normocholesterolemic			Hypercholesterolemic			*p* Value		
	OPO *n* = 65	HOSO *n* = 34	SO *n* = 31	OPO *n* = 67	HOSO *n* = 30	SO *n* = 37	Oil	N/H	Oil × N/H
**Weight**	−0.3 ± 0.1	0.2 ± 0.2	0.0 ± 0.2	0.0 ± 0.1	0.1 ± 0.2	0.1 ± 0.2	0.392	0.508	0.715
**BMI**	−0.4 ± 0.2	0.3 ± 0.5	0.3 ± 0.5	0.1 ± 0.1	0.0 ± 0.3	0.4 ± 0.3	0.251	0.700	0.421
**Body fat**	−0.3 ± 1.5	1.8 ± 2.3	−1.0 ± 1.1	2.5 ± 1.5	3.3 ± 3.0	3.8 ± 2.2	0.476	0.064	0.726
**Visceral fat**	−1.5 ± 2.4	−3.9 ± 2.6	2.2 ± 4.0	−1.1 ± 1.3	−2.9 ± 2.4	6.6 ± 6.1	0.146	0.706	0.953
**Waist**	−0.5 ± 0.5	1.1 ± 0.9	−0.8 ± 0.9	−0.8 ± 0.4	1.6 ± 1.3	−0.6 ± 0.8	0.026	0.745	0.620
**Hip**	0.6 ± 0.9	1.1 ± 0.4	0.4 ± 0.4	0.3 ± 0.2	0.0 ± 0.2	1.2 ± 0.9	0.619	0.615	0.335

Values represent mean ± standard error of mean. The rate of change was calculated from [(post-treatment value − pre-treatment value)/pre-treatment value] and expressed as percentage. A linear mixed model was used for data analysis. *p*-value in the first column represents the treatment effect (OPO, HOSO or SO), in the second column the group effect [normocholesterolemic (N) and hypercholesterolemic (H)], and in the last column the interaction of treatment and group. Significance level was set at *p* < 0.05. BMI: body mass index.

**Table 5 foods-11-02186-t005:** Effects of olive pomace oil (OPO), high oleic sunflower oil (HOSO) and sunflower oil (SO) on oxidation and antioxidant biomarkers.

(%)	Normocholesterolemic			Hypercholesterolemic			*p* Value		
	OPO *n* = 65	HOSO *n* = 34	SO *n* = 31	OPO *n* = 67	HOSO *n* = 30	SO *n* = 37	Oil	N/H	Oil × N/H
**ABTS (µM TE)**	11.4 ± 5.8	7.2 ± 3.5	7.6 ± 6.3	2.1± 3.4	8.0 ± 2.6	2.4 ± 3.8	0.284	0.987	0.263
**ORAC (µM TE)**	3.7 ± 3.0	1.1 ± 6.0	−0.8 ± 3.6	2.9 ± 3.3	18.8 ± 10.7	1.4 ± 3.6	0.233	0.210	0.321
**FRAP (µM TE)**	4.0 ± 1.7	7.6 ± 3.4	2.7 ± 2.6	2.1 ± 1.8	1.5 ± 2.8	−2.6 ± 1.6	0.276	0.018	0.469
**oxLDL (ng/mL)**	15.0 ± 7.8	10.5 ± 9.4	1.5 ± 7.6	5.5 ± 9.7	16.4 ± 11.3	2.5 ± 5.1	0.418	0.746	0.478
**MDA (nmol/mL)**	−4.4 ± 2.2 ^a^	3.1 ± 3.2 ^b^	0.7 ± 2.9 ^ab^	0.1 ± 2.0	−8.4 ± 4.7	−1.4 ± 2.9	0.739	0.140	0.004

Values represent mean ± standard error of mean. The rate of change was calculated from [(post-treatment value − pre-treatment value)/pre-treatment value] and expressed as percentage. A linear mixed model was used for data analysis. *p*-value in the first column represents the treatment effect (OPO, HOSO or SO), in the second column the group effect [normocholesterolemic (N) and hypercholesterolemic (H)], and in the last column the interaction of treatment and group. Significance level was set at *p* < 0.05. ABTS: free radical 2,2′-azinobis-(3-ethylbenzothiazoline-6-sulfonic acid)-scavenging capacity. ORAC: oxygen radical absorbance capacity. FRAP: ferric reducing/antioxidant power. MDA: malondialdehyde. TE: Trolox equivalents. OxLDL: Oxidised low density lipoprotein. Superscripts correspond to significant differences within the normocholesterolemic (N) or hypercholesterolemic (H) group according to the Bonferroni test.

## Data Availability

The data presented in this study are available on request from the corresponding author. The data are not publicly available due to privacy concerns.

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
