# Peer review of "Olive Pomace Oil versus High Oleic Sunflower Oil and Sunflower Oil: A Comparative Study in Healthy and Cardiovascular Risk Humans"

_foods, 2022, doi:10.3390/foods11152186_

Round 1
Reviewer 1 Report
The presented manuscript discusses about interesting aspects of possible improvement of human health, primarily about the cardiovascular risk by ingesting olive pomace oil. The present study includes two clinical trials comparing olive oil pomace intake with widely used sunflower oil and nutritionally improved high oleic sunflower oli in a population of normocholesterolemic and hypercholesterolemic subjects.
This research covered a lot of aspects, the idea is innovative and well developed, the analyzes are appropriate. The paper is generally well but slightly confusingly written, and requires corrections or clarifications according to the following comments:
Headings as background, methods etc. should not listed in the abstract.
Line 15-16, Only two human studies ...unclear, rearrange this sentence
Line 93, The inscription for figure one should be below the picture
Table 1. includes results and should be move from material and methods section to results and discussion section, as well as a discussion related to Table 1.
Line 166, include Friedewald formula to be available to reader, not just a reference.
Line 187-189, Which methodology was used for ABTS, ORAC and FRAP assays? Same as the authors previously developed in references provided?
Line 203-213, Whether the authors determined the chemical composition of the oils or whether these data were taken from other authors? If so, write in more detail about the chemical composition methodology and obtained results present in the results and discussion section.
Line 216- 224, This part should not stand after the heading - statistical method. Rearrange it and move to the the last paragraph of the introduction.
Line 256-263, Provide citations.
Table 2., Put caption above the table, Check table- unclear.
Emphasize the ecological significance of use olive pomace oil through discussion, too.
In conclusion the authors could present their suggestions on this subject or suggestions for future research.
Author Response
The authors would like to thank the editor and referees for their revision. We feel that their comments have greatly contributed to improve our work. Track-changes have been used to easily identify the modifications made. In addition, the new sentences added have been shaded in yellow (referee 1), green (referee 2) and grey (referee 3).

Reviewer 2 Report
Comments to the manuscript foods-1804320 Olive pomace oil versus high oleic sunflower oil and sunflower oil: a comparative study in healthy and cardiovascular risk humans.
The manuscript is the report of two experiments of comparison of olive pomace (OPO) supply to two groups of humans characterized as normocholesterolemic and hypercholesterolemic. The OPO supply is alternated with sunflower oil (SO) or high oleic sunflower oil (HOSO). Two periods of oil supply of 4 weeks are preceded by two periods of Run-in and Whash-out. During the Run-in period all the participants eated SO, but during the whash-out weeks the participants eated SO (OPO and HOSO treatments) and corn oil (SO treatment). This difference in the participant treatments made not comprable the data obtained in the two experiments, as statistically performed in the manuscript. Moreover, the corn oil used was not chemically characterized and described in the manuscript. In my opinion, the manuscript is not acceptable in the present form and Authors should divide the results of the two esxperiments providing a separate discussion of the results obtained. The present data treatment is not statistically sound.
Author Response

(The authors gave the same response as above.)

Reviewer 3 Report
The study is interesting however I have the follwoing querries:
1) only one sample per category has been used, so how representative were in terms of composition so that the findings to be not just random ?
2) I would recommend to the authors in order to improve discussion and highlight better the findings to consult and include in the reference list the EC regulations:
REGULATION (EC) No 1924/2006 OF THE EUROPEAN PARLIAMENT AND OF THE COUNCIL of 20 December 2006 on nutrition and health claims made on foods
COMMISSION REGULATION (EU) No 432/2012 of 16 May 2012 establishing a list of permitted health claims made on foods, other than those referring to the reduction of disease risk and to children’s development and health
REGULATION (EU) No 1169/2011 OF THE EUROPEAN PARLIAMENT AND OF THE COUNCIL of 25 October 2011 on the provision of food information to consumers, amending Regulations (EC) No 1924/2006 and (EC) No 1925/2006 of the European Parliament and of the Council, and repealing Commission Directive 87/250/EEC, Council Directive 90/496/EEC, Commission Directive 1999/10/EC, Directive 2000/13/EC of the European Parliament and of the Council, Commission Directives 2002/67/EC and 2008/5/EC and Commission Regulation (EC) No 608/200
To assist, I highlight the following excerpts
Vitamin Ε: Vitamin E contributes to the protection of cells from oxidative stress. The claim may be used only for food which is at least a source of vitamin E as referred to in the claim SOURCE OF [NAME OF VITAMIN/S] AND/OR [NAME OF MINERAL/S] as listed in the Annex to Regulation (EC) No 1924/2006.
SOURCE OF (NAME OF VITAMIN/S) AND/OR (NAME OF MINERAL/S) A claim that a food is a source of vitamins and/or minerals, and any claim likely to have the same meaning for the consumer, may only be made where the product contains at least a significant amount.
HIGH (NAME OF VITAMIN/S) AND/OR (NAME OF MINERAL/S) A claim that a food is high in vitamins and/or minerals, and any claim likely to have the same meaning for the consumer, may only be made where the product contains at least twice the value of ‘source of (NAME OF VITAMIN/S) and/or (NAME OF MINERAL/S)’
ANNEX XIII REFERENCE INTAKES [(PART A — DAILY REFERENCE INTAKES FOR VITAMINS AND MINERALS (ADULTS) 1. Vitamins and minerals which may be declared and their nutrient reference values (NRVs)]: Vitamin E (mg)= 12
Significant amount of vitamins and minerals: As a rule, the following values should be taken into consideration in deciding what constitutes a significant amount: 15 % of the nutrient reference values specified in point 1 supplied by 100 g or 100 ml in the case of products other than beverages.
Oleic acid: Replacing saturated fats in the diet with unsaturated fats contributes to the maintenance of normal blood cholesterol levels. Oleic acid is an unsaturated fat. The claim may be used only for food which is high in unsaturated fatty acids, as referred to in the claim HIGH UNSATURATED FAT as listed in the Annex to Regulation (EC) No 1924/2006.
Monounsaturated and/or polyunsaturated fatty acids: Replacing saturated fats with unsaturated fats in the diet contributes to the maintenance of normal blood cholesterol levels [MUFA and PUFA are unsaturated fats] The claim may be used only for food which is high in unsaturated fatty acids, as referred to in the claim HIGH UNSATURATED FAT as listed in the Annex to Regulation (EC) No 1924/2006
HIGH MONOUNSATURATED FAT: A claim that a food is high in monounsaturated fat, and any claim likely to have the same meaning for the consumer, may only be made where at least 45 % of the fatty acids present in the product derive from monounsaturated fat under the condition that monounsaturated fat provides more than 20 % of energy of the product.
HIGH UNSATURATED FAT: A claim that a food is high in unsaturated fat, and any claim likely to have the same meaning for the consumer may only be made where at least 70 % of the fatty acids present in the product derive from unsaturated fat under the condition that unsaturated fat provides more than 20 % of energy of the product.
Author Response

(The authors gave the same response as above.)

Round 2
Reviewer 2 Report
I appreciate the efforts of the Authors to perform the changes required by the reviewers. Even if I confirm my criticism regarding the use of the of the corn oil in the Wash-Out period, in my opinion the manuscript is now acceptable for publication after the minor editing changes.
Author Response
Thank you for your comments. We have replied in the file attached.

Reviewer 3 Report
Although i still beleive that the composition of the oils should have been discussed with regards to the ranges in literature or an average one, I think the mmanuscript has been improved and i suggest publication
Author Response

(The authors gave the same response as above.)
